# Understanding Mechanisms of RKIP Regulation to Improve the Development of New Diagnostic Tools

**DOI:** 10.3390/cancers14205070

**Published:** 2022-10-17

**Authors:** Massimo Papale, Giuseppe Stefano Netti, Giovanni Stallone, Elena Ranieri

**Affiliations:** 1Unit of Clinical Pathology, Department of Laboratory Diagnostics, University Hospital “Policlinico Foggia”, 71122 Foggia, Italy; 2Unit of Clinical Pathology, Center for Molecular Medicine, Department of Medical and Surgical Sciences, University of Foggia, 71122 Foggia, Italy; 3Unit of Nephology, Dialysis and Transplantation, Advanced Research Center on Kidney Aging (A.R.K.A.), Department of Medical and Surgical Sciences, University of Foggia, 71122 Foggia, Italy

**Keywords:** RKIP, PhosphoRKIP, cancer, biological fluids, tissue, biomarkers

## Abstract

**Simple Summary:**

Raf Kinase Inhibitor protein is a protein that governs multiple intracellular signalling involved primarily in the progression of tumours and the development of metastases. In this review, we discussed the main mechanisms that regulate the expression and activity of RKIP with the aim of identifying the link between the transcriptional, post-transcriptional and post-translational events in different tumour settings. We also tried to analyse the studies that have measured the levels of RKIP in biological fluids in order to highlight the possible advantages and potential of RKIP assessment to obtain an accurate diagnosis and prognosis of various tumours.

**Abstract:**

One of the most dangerous aspects of cancer cell biology is their ability to grow, spread and form metastases in the main vital organs. The identification of dysregulated markers that drive intracellular signalling involved in the malignant transformation of neoplastic cells and the understanding of the mechanisms that regulate these processes is undoubtedly a key objective for the development of new and more targeted therapies. RAF-kinase inhibitor protein (RKIP) is an endogenous tumour suppressor protein that affects tumour cell survival, proliferation, and metastasis. RKIP might serve as an early tumour biomarker since it exhibits significantly different expression levels in various cancer histologies and it is often lost during metastatic progression. In this review, we discuss the specific impact of transcriptional, post-transcriptional and post-translational regulation of expression and activation/inhibition of RKIP and focus on those tumours for which experimental data on all these factors are available. In this way, we could select how these processes cooperate with RKIP expression in (1) Lung cancer; (2) Colon cancer, (3) Breast cancer; (4) myeloid neoplasm and Multiple Myeloma, (5) Melanoma and (6) clear cell Renal Cell Carcinoma. Furthermore, since RKIP seems to be a key marker of the development of several tumours and it may be assessed easily in various biological fluids, here we discuss the potential role of RKIP dosing in more accessible biological matrices other than tissues. Moreover, this objective may intercept the still unmet need to identify new and more accurate markers for the early diagnosis and prognosis of many tumours.

## 1. Overview on Raf Kinase Inhibitor Protein (RKIP) and Its Biological Functions

Raf kinase inhibitory protein (RKIP) is the endogenous inhibitor of Raf Kinase, a family of three serine/threonine-specific protein kinases that activate MAP kinase (MAPK) signalling. The importance of the map kinase pathway in the regulation of key cellular processes such as proliferation, differentiation and development clearly highlights the impact of its regulation to preserve normal cell functions [1].

RKIP is widely expressed in normal human tissues, where it modulates many physiological processes, such as spermatogenesis, neural development, cardiac output and membrane biosynthesis [2,3].

Furthermore, given the centrality of MAP Kinase in the progression of many neoplasms, there is a growing interest in understanding and modulating the mechanisms involved in its regulation which may be used as novel targets for successful therapy. 

RKIP regulates the activation of MAP kinase through binding to Raf-1, which prevents its phosphorylation. At the same time, the phosphorylation of RKIP by Protein Kinase C induces the release of Raf and, consequently, the activation of the MAP kinase pathway (Figure 1).

Of note, RKIP activation/inactivation can indirectly interfere with upstream activators of Raf-1, such as G-protein coupled receptors (GPCR). In fact, RKIP phosphorylation by protein kinase C (PKC) not only releases (and activates) Raf-1 but also associates RKIP with a GPCR inhibitor namely GRK2. In this state GPCR may further stimulate Raf-1 phosphorylation and activation thus, enhancing the MAP Kinase pathway [4] (Figure 1).

Interestingly, RKIP can modulate other signalling pathways such as those related to nuclear factor kappa-light-chain-enhancer of activated B cells (NFκB) and glycogen synthase kinase 3 beta (GSK3β), thus exerting a key role in regulating the inflammation process [5].

NF-κB is a pro-inflammatory transcription factor that is activated by a number of several cytokines in inflammatory cells. It remains in an inactive form in the cytoplasm through association with inhibitory kappa B (IκB). RKIP inhibits NF-κB signalling through interaction with the upstream kinases (NIK, TAK, IKKα and IKKβ), thus inducing the inhibition of IκBα phosphorylation and the accumulation of NF-κB in the cytoplasm [6,7,8] (Figure 1).

Recently, Al-Mulla et al. reported that RKIP may serve also as an activator of GSK3β that acts as a downstream regulatory switch for numerous signalling pathways. Among them, RKIP is involved in glycogen metabolism, cell development, gene transcription, and protein translation to the cytoskeletal organization, cell cycle regulation, proliferation, and apoptosis [9].

RKIP is physically associated with GSK3β, and its overexpression suppresses the phosphorylation of GSK3β at residue 390 leading to GSK3β inactivation. RKIP downregulation would otherwise promote phosphorylation at this inhibitory site and lead to the stabilization of the GSK3β protein that, in turn, may promote gene transcription and cell proliferation (Figure 1).

In the next paragraphs, the state of the art of mechanisms involved in the regulation of RKIP expression and activity will be discussed. Moreover, the correlation of RKIP and pRKIP levels with the onset and progression of many cancer types as well as the potential clinical use of RKIP concentration assessment in various biological matrices as a surrogate marker for diagnosis, prognosis and therapy will be also reported.

## 2. Mechanisms of RKIP Regulation

RKIP expression is regulated mainly at transcriptional and post-transcriptional levels, which include promoter methylation and post-transcriptional regulation by microRNAs or degradation. Results coming from various studies indicate that epigenetic regulation of RKIP expression is a complex process. It results from a mechanism of histone acetylation or methylation and involves multiple factors in a cell- or tissue-dependent manner [10]. At the epigenetic level, the RKIP promoter is frequently found methylated in most cancers. The importance of hyper-methylation for RKIP silencing has been reported in gastric adenocarcinomas, colorectal cancer, breast cancer and oesophageal squamous cell carcinomas [11,12,13,14] (Figure 1). Recently, Zaravinos et al. correlated CpG site methylation and gene expression in melanomas dataset [15]. The data, extracted from the TCGA-SKCM dataset (https://portal.gdc.cancer.gov/projects/TCGA-SKCM, accessed on 12 September 2022) indicated an inverse correlation between RKIP mRNA expression and PEBP1 methylation probe (cg00091483) in position 118574757 of chr12 thus, providing clear evidence that CpG island hyper-methylation silences RKIP expression. However, the correlation between the hyper-methylation of RKIP promoters and clinical outcomes may not always be clear. In fact, while studies on oesophageal cancer correlate RKIP hyper-methylation with lower tumour differentiation, the acquisition of a metastatic phenotype, and an overall reduction of patient survival, it does not seem to predict patients’ survival in breast cancer cohorts. This could suggest that epigenetics is only one of the mechanisms that contribute to regulating RKIP expression and that its impact could depend on the characteristics of the tumour and its microenvironment.

Furthermore, the expression of RKIP is finely regulated by various Transcription Factors (TF) that bind its promoter and determine the expression in physiological and pathological conditions. TFs may act as positive or negative regulators of RKIP expression. Among positive regulators, there are cAMP-dependent transcription factors CREB, Sp1 and histone acetylase p300 [16]. In vitro studies on melanoma and cervical cancer cells showed, in fact, a direct correlation between CREB, Sp1 and p300 intracellular levels and RKIP expression. Other TFs such as the Androgen receptor (AR) may activate, after androgen stimulation, prostate-specific RKIP expression in certain forms of prostate cancer [17].

Another group of TFs exerts an inhibitory effect. They include SNAIL and BACH1 whose direct binding to the RKIP promoter suppresses RKIP transcription and expression in prostate [18] and breast cancer [19] (Figure 2).

At the molecular level, SNAIL represses RKIP transcription by binding to E-boxes in the RKIP promoter and by recruiting Sin3A/histone deacetylases (HDACs), histone methyltransferase polycomb group protein (PcG) and a multicomponent protein complex named PRC2 [20]. On the whole, this multiprotein complex inhibits the binding of a transcription factor for RNA polymerase IID (TFIID) and impairs RKIP expression [21].

BACH1, a basic leucine zipper transcription factor that regulates oxidative stress and stress-induced senescence, is another inhibitory TF that exerts its inhibitory mechanism by binding a specific site located at 173 bp from the transcription initiation site and by recruiting EZH2, a component of PRC2, to the RKIP promoter [19].

It is worth noting that, RKIP also inhibits BACH1 by targeting the Erk1/2-myc-let7 signalling pathway [22], therefore RKIP and BACH1 act by regulating each other in a mechanism whereby the increase of one determines the suppression of the other and vice versa. In this context, RKIP and BACH1 cooperate, in cancer cells, to the onset of two stable states namely, an anti-metastatic state with high RKIP-low BACH1 and a pro-metastatic state with low RKIP-high BACH1 expression [19].

Furthermore, RKIP can interfere with SNAIL activation through various mechanisms including inhibition of the TF NF-κB [23,24,25]. Therefore, SNAIL and RKIP are activated or repressed through a double regulatory circuit similar to that described for RKIP and BACH1.

RKIP expression in prostate [18] and breast cancer [19] is initiated primarily by binding of the transcription factors BACH1 and SNAIL 1 to the RKIP promoter which suppresses RKIP transcription and expression (Figure 2). Various studies have highlighted the link between the expression of BACH1 and SNAIL1, the downregulation of RKIP and the onset of Epithelial-Mesenchymal Transitions (EMT) [18,22,26,27]. These findings have been corroborated, in breast cancer cohorts, by finding that downregulation of RKIP appears to be proportionally related to increased expression of SNAIL 1, BACH1 and some other EMT markers such as vimentin and Zinc Finger E-Box Binding Homeobox 1/2—ZEB1/2 [10]. Although this evidence may suggest a direct implication of RKIP in the EMT, most of the studies have been carried out on cellular models, hence it has not yet been possible to mechanistically establish the direct link between inhibition of RKIP by SNAIL 1 and BACH1, and the expression of mesenchymal-epithelial transition markers. At the post-transcriptional level, RKIP mRNA is targeted by a number of micro RNAs that act as inhibitors of RKIP expression in many cancer types (Figure 2). For example, miR-543 [28] and miR-23a have been reported as suppressors of RKIP expression in prostate cancer [11,28], miR-27a in lung cancer [29] and miR-224 in breast cancer [30]. Of note, the main mechanism that regulates the association or dissociation of RKIP from its targets is post-translational modifications, particularly phosphorylation. Serine 153 phosphorylation (pSer153-RKIP) by protein kinase C ζ (PKCζ) has been found in some cancer types to account for the loss of RKIP activity that in turn activates MAP kinase pathway and inhibit GPCR through inhibition of G protein-coupled receptor kinase 2 (GRK2). Furthermore, in some cancers, the nuclear pSer153-RKIP levels significantly correlate with poor response to therapy and overall prognosis [31,32] (Figure 2). Furthermore, the downregulation of RKIP may promote the deregulation of major cellular mechanisms such signalling, mechano-sensing, autophagy and migration [33]. Specifically, a direct interaction of RKIP with specific targets implicated in cell proliferation and survival (GRK2), migration and invasion (IQGAP) and autophagy (LC3 and Rab8) have been reported. Recent interactome analyses suggest that RKIP may act as a cellular organizer of the response to various stimuli by acting on cytoskeleton reorganization and membrane remodelling [34].

## 3. RKIP and Cancer

Most of the molecular events that determine the downregulation of RKIP have been analysed in the previous paragraphs. These events, common to many tumours, increase the invasiveness and ability to form metastases. In this scenario, the assessment of basal levels of RKIP can constitute an important prognostic index. As reported [32], low levels of the protein can depend as much on a transcriptional regulation as on a post-translational regulation, thus we will discuss the weight of one or the other factor in the context of the main neoplasms. Most of the studies available up to now have evaluated the expression levels of biomarker candidates, especially from the transcriptomic point of view [35,36,37,38]. Protein levels were mainly evaluated by tissue microarray while only a limited number of studies evaluated post-translational changes [39]. Although RKIP is expressed in almost all tissues of the body, the levels of expression vary from one tissue to another. The data reported in the Human Protein Atlas Project (https://www.proteinatlas.org/ENSG00000089220-PEBP1, accessed on 12 September 2022) show that, at the transcriptomic level, RKIP is mainly expressed in liver, adrenal gland, and kidney. The central nervous system, intestine and skeletal and cardiac muscle show average levels of expression while basal RKIP is gradually less expressed in other organs such as the prostate, breast and lung. Protein expression of RKIP in various tissues is roughly in line with the gene expression. Indeed, the analysis of various datasets suggests an increased expression in the thyroid, parathyroid, liver, adrenal and kidney. The central nervous system, lung, digestive tract, pancreas, breast, prostate and heart muscle show average protein levels while other organs such as the colon, bladder and skeletal muscle have lower levels. 

In the next paragraphs, we will discuss only those tumours for which RKIP has been investigated at transcriptomic, proteomic, and post-translational levels. The salient aspects of the role of RKIP in each tumour type are summarized in Table 1.

### 3.1. Lung Cancer

Non-small cell lung cancer (NSCLC) accounts for 80–90% of lung cancer cases and is among the most malignant types of cancer also because 50–70% of these patients are diagnosed at the advanced stage of disease [40,41].

RKIP mRNA expression levels are significantly downregulated in NSCLC and correlate with poorer differentiation and advanced tumour-node-metastasis stage [42].

Wang et al. recently reported that the differences in RKIP protein levels were statistically significant between patients with early-stage (TNM I/II) versus late-stage (stage III/IV) NSCLC [43] with reduced RKIP expression levels in patients with late-stage NSCLC.

These results are consistent with those recently reported by Meng et al. [44] on 156 NSCLC patients where, interestingly, lower RKIP expression predicted worse overall survival in stage I and II patients (*p* = 0.011, log-rank) but not in stage III and IV patients (*p* = 0.711, log-rank). Furthermore, lower levels of RKIP in primary NSCLC may constitute a negative prognostic index also for the response to radiotherapy [45].

NSLCL progression is closely associated with both the expression of miRNAs and long non-coding RNAs (LcnRNAs). Luo D et al. [46] reported the role of miR-362 in the development of NSCLC metastatic cells while Wang et al. [47] demonstrated that miR-543 overexpression exerts tumour-promoting effects via repressing PTEN. Recently, a number of long non-coding RNAs (LcnRNAs) have been described in association with the prognosis of lung cancer [48]. LcnRNAs modulate gene transcription or translation through epigenetic regulation, RNA splicing, chromatin remodelling, and microRNA sponging (miRNA). Among lncRNAs, GATA6 antisense RNA 1 (GATA6-AS1) is often down-regulated in lung squamous cell carcinoma and correlated with poor prognosis [48]. Recent studies show an inverse correlation between the expression levels of GATA6 antisense RNA 1 and those of microRNA-543. In this model, the reduction of GATA6 antisense RNA 1, often found in NSCLC, would be indirectly responsible for tumour progression by increasing microRNA-543 levels that, in turn, suppresses RKIP expression [49]. 

Furthermore, in NSLC RKIP downregulation has been associated with the dysregulation of the miR-150-FOXO4 axis that may promote EMT through the NF-κB/SNAIL/YY1/RKIP loop [50]. Given the importance of RKIP in the regulation of many intracellular signalling, it is not surprising that its downregulation can have a significant impact on critical processes for tumour progression such as EMT, angiogenesis and cell proliferation [51]. Briefly, RKIP regulates, directly or indirectly, many signalling pathways by inhibiting the impaired activation of TF. For example, it inhibits the translocation of NICD to the nucleus and the subsequent activation of EMT genes involved in metastases and progression. Furthermore, it prevents the unbalanced activation of Raf/MEK/ERK/STAT3 signalling that is implicated in angiogenesis, proliferation and metastatization as well as blocking SNAIL through MAPK inhibition and NF2 stabilization. Finally, RKIP binds to the SMO receptor, keeping it inactive and preventing the transcription of Gli1, which acts as a transcriptional activator of numerous genes, regulating proliferation, differentiation, extracellular matrix interactions, and cancer stem cell (CSC) activation.

Data on the role of post-translational changes in lung cancer progression are still very scarce. In fact, only the study conducted by Huerta-Yepez S et al. [52] analysed protein levels of RKIP and pRKIP in a large cohort of lung cancer patients. One of the limitations of this study was the heterogeneity of the sample which included both adenocarcinomas and Squamous Cell Carcinomas. The authors described a statistically significant reduction in the phosphorylated form of RKIP between metastatic and non-metastatic tumours. This data appears to contradict the function of RKIP as the reduction of pRKIP should increase the active fraction of the protein with an overall protective effect towards tumour progression. 

Hence, to understand and explain the results obtained by Huerta-Yepez and co-workers, it would be significant to clarify at what stage of tumour development pRKIP was assessed. In fact, at a very early stage, a transient increase of pRKIP could be plausible as it might reflect the body’s attempt to eliminate tumour cells by an increased inflammatory response. This could be in line with the finding of Albano and co-workers [53] who described increased phosphorylation of pulmonary RKIP in long-term exposure to proinflammatory and oxidative stimuli like cigarette smoking and chronic obstructive pulmonary disease. 

On the contrary, during tumour progression, the reduction of pRKIP could represent an epiphenomenon linked to the overall downregulation of RKIP in tumour tissue. In fact, pRKIP is only a small percentage of the total RKIP thus pRKIP downregulation may be an earlier effect on the downregulation of RKIP. In this scenario, the reduction of phosphorylated RKIP precedes the reduction of the whole protein and its assessment could be useful as an early marker of tumour development.

### 3.2. Colon Cancer 

Cytoplasmic RKIP expression has been reported in normal colonic mucosa [54]. Specifically, RKIP appears undetectable in the crypts but steadily increases as cells move upward and differentiate. It is also expressed in the ganglia of Auerbach’s myenteric plexus, and in chromogranin A—positive neuroendocrine cells [55].

The rate of expression of RKIP in primary colon cancer correlates with the development of metastases and may predict overall survival. In fact, reduced RKIP expression in primary CRC predicts metastatic recurrence and disease-free survival (91 versus 61 months respectively in patients with positive or weak/negative RKIP staining). Furthermore, RKIP expression is associated with the degree of differentiation of colorectal cancer cells since well-differentiated cell lines show higher expression than poorly differentiated cell lines [56]. Thus, the evaluation of RKIP expression in primary Colorectal cancer (CRC) can be useful for identifying early-stage CRC patients at risk of relapse [55]. In this context, the loss of RKIP inhibits cell cycle arrest and promotes cell proliferation making it not only a metastasis inhibitor factor but also a key player in colorectal cancer cell differentiation. 

The analysis of the mechanisms of regulation of RKIP expression in colon cancer allowed for the identification of a significant correlation with some miRNAs.

Oberg et al. [57] compared miRNAs expression in colorectal cancers, matched to healthy tissue and colorectal adenomas with low-grade and high-grade dysplasia to identify miRNAs that may promote colorectal cancer progression. Among differentially expressed miRNAs they found miR-224, a miRNA already described as an early predictor of colorectal cancer development [57,58]. Of note, increased miR-224 activates RAS/MAPK signalling, either through activation of KRAS or promotion of RAF1/MEK binding following RKIP activation, thus suggesting a key role of the RKIP network in promotion of colon cancer development as a result of the alteration of miR-224. Another miRNA implicated in CRC development and progression is miR-330. Shirjang et al. [59] described a significant downregulation of miR-330-3p and miR330-5p in CRC tumours compared to normal controls. Further, induction of miR-330 may prevent the proliferation of CRC cells by reducing BACH1 expression, which, in turn, represses MMP9, C-X-C chemokine receptor type 4 (CXCR4), and vascular endothelial growth factor receptor (VEGFR). As already reported [16,18,19,20], a number of studies have highlighted the link between the expression of the transcription factors BACH1 and SNAIL1, the downregulation of RKIP and the onset of Epithelial-Mesenchymal Transitions (EMT), therefore the downregulation of miR330 may sustain the EMT at least in part through the dysregulation of the RKIP network.

If the analysis of the expression levels of RKIP and that of the epigenetic factors involved in its regulation have been well described by various authors, the role of post-translational modifications is still largely unexplored. Only one study has investigated the role of pRKIP in colorectal cancer so far [31]. It is worth noting an interesting correlation between the distribution of the phosphorylated form of RKIP and the degree of invasiveness of colorectal cancer. The authors observed a shift of pRKIP from the cytoplasm to the nucleus in the process of tumour progression, in fact, patients with worse prognosis, namely high tumour grade and lymph node infiltration, had reduced cytoplasmic pRKIP and conversely increased nuclear pRKIP. Of note, there were no statistically significant associations between nuclear and cytoplasmic RKIP and the grade or lymph node infiltration. These data, if confirmed, could indicate that the distribution of pRKIP between the nucleus and cytoplasm of cancer cells may constitute an early event in the process of colorectal cancer progression and makes the evaluation of pRKIP potentially useful to obtain a more appropriate classification of patients.

### 3.3. Breast Cancer

RKIP is physiologically poorly expressed in breast tissues. The data reported in the Human Proteome Atlas show that both at the mRNA and protein level, RKIP is expressed mainly in glandular and myoepithelial cells while it is not detectable in adipocytes. 

In aggressive metastatic breast cancer, RKIP expression is repressed mainly at the transcriptional level by the BACH1/Snail/EZH2 repressor complex through the molecular mechanisms previously reported [18,22,60]. Furthermore, a direct correlation between the expression levels of specific microRNAs and those of RKIP has been described.

Huang L et al. [30] reported that miR-224 inhibits RKIP expression in highly invasive breast cancer cell lines through the binding of its 3′-untranslated region (3′-UTR) that, in turn, leads to the upregulation of CXCR4, MMP1, and OPN, which are involved in breast tumour metastasis to the bone. Further, Zou Q. et al. [61] reported that the reduction of RKIP and miR-185 induces the expression of HMGA2, a non-histone protein binding to chromatin which is closely related to tumourigenesis, invasion and metastasis of tumours. In particular, the authors showed that lower levels of RKIP in metastatic breast tumours depress the expression of the inhibitory microRNA miR185 and, consequently, increase HMGA2 that, in turn, sustains malignancy and invasiveness of cancer cells. These results are in line with those reported by Yun et al. [18] who demonstrated RKIP’s ability to reduce the expression of BACH1 and HMGA2 through direct regulation of let-7. Low levels of RKIP expression in breast cancer, initially described in cellular models, have also been confirmed by immunohistochemical studies in patients’ tissues. For example, Al Mulla et al. [62] analysed RKIP expression levels in tissues of breast cancer patients and described a specific association between RKIP levels and tumour size and grade. In addition, the expression levels of RKIP made it possible to predict the future development of metastases, confirming the importance of this protein in the metastatic process. These authors also investigated the role of pRKIP by evaluating pRKIP expression levels in 373 cases of breast cancer. After stratifying the population into 4 groups (low, medium, medium-high and high) according to the levels of pRKIP expression in the nucleus and cytoplasm, they correlated pRKIP levels with the clinicopathological parameters of the patients. Although no statistically significant correlation was found with parameters commonly used to predict tumour severity (e.g., stage, grade, size, and metastasis to lymph nodes), patients with higher pRKIP levels showed about 20 months longer survival than patients with lower pRKIP levels. In addition, low levels of pRKIP correlated with a greater likelihood of developing a recurrence. Combining data from various studies, it can be hypothesized that the reduction of RKIP by epigenetic and post-transcriptional mechanisms is a negative prognostic factor while the transition of RKIP from the native to the phosphorylated form could constitute a protective factor against tumour progression, metastasis and death.

### 3.4. Myeloid Neoplasms (MNs) and Multiple Myeloma 

CD34+ hematopoietic stem and progenitor cells (HSPCs) significantly express RKIP. Nevertheless, its expression is very reduced in differentiated myeloid leukocytes, including granulocytes and monocytes, but not in lymphocytes that show an expression comparable to the HSPC compartment [63]. RKIP acts as a tumour suppressor in primary Acute Myeloid Leukaemia (AML) its loss increases the invasion and migration potential of a series of AML cell lines and promotes, in vivo, the formation of extramedullary metastases [64]. Of note, Hatzl et al. [65] analysed up to 400 AML patients and demonstrated a direct correlation between increased expression of miR-23a and the downregulation of RKIP in this cohort. They reported that miR-23a directly targets the 3′UTR of RKIP, which, in turn, causes RKIP downregulation in hematopoietic cells. There is currently no literature data on the role of phosphorylated RKIP in AML. However, given the importance of this process in the inactivation of RKIP, it is very likely that this role will be investigated soon as done in multiple myeloma (MM), a plasma-cell neoplastic disorder arising from a premalignant disease known as monoclonal gammopathy of undetermined significance (MGUS). 

Baritaki S. et al. initially reported, in MM tissue compared to normal bone marrow cells, a direct correlation between higher expression of RKIP and YY1, a transcription factor implicated in the modulation of tumour cell chemo/immuno-resistance [66]. In these cells, YY1 seems to concentrate mainly in the nucleus while in the cells of the healthy bone marrow this transcription factor is more present in the cytoplasm. 

Further studies clarified that the higher concentration of nuclear YY1, observed in patients with progressive disease, favours the activation of chemo/immune-resistance mechanisms regulated by this transcription factor [67].

It is worth noting that most RKIP, increased in both MM cell lines and patient-derived tumour tissues compared to healthy B cells, and healthy bone marrow has been further characterized as phosphorylated-RKIP [66]. 

This study was the first experimental evidence of how the phosphorylation of RKIP can mimic its downregulation, determining, in fact, the same effect observed in other neoplasms. In fact, in most tumours, RKIP is mainly downregulated, while in MM it is mainly imbalanced by a significant increase of inactive phosphorylated form that stimulates intracellular signalling mediated by MAPK and sustains cell proliferation and invasiveness. 

### 3.5. Melanoma 

Primary lesions of early-stage malignant melanoma show an overall decrease of RKIP when compared with benign lesions (i.e., nevi) [68]. As RKIP downregulation alters cellular processes closely related to the neoplastic transformation such as proliferation and migration it might be a useful marker for early diagnosis rather than prognosis. In fact, experimental data indicate that RKIP expression levels are not different among primary (Mel-HO, A375) and metastatic (HT-144, Hs-294T, Colo-800) cell lines neither between stage I–II patients with a favourable or unfavourable prognosis. Of note, Cardile et al. [69] found significant increases in pRKIP in non-metastatic melanoma (Clark I–II) with respect to control and metastatic melanomas thus combined analysis of RKIP and pRKIP might be a useful tool to obtain early diagnosis and prognosis. Mechanistically, RKIP acts primarily as tumour suppressor also in melanoma, in fact, its silencing results in increased expression of various oncogenes (KIT, BCL3, MAF, MYC, MYCL, HOXA9, CDC25B, and PIM1), most of which are associated with the transcription factor NANOG. NANOG is a known inducer of a stem cell-like state that has been found aberrantly expressed in many kinds of tumours [70,71]. In melanoma, melanosphere formation correlates with increased expression of NANOG which induces the transcription of genes regulating motility and tissue transmigration as well as favours SNAIL-1-mediated activation of the key genes involved in EMT [72]. Furthermore, post-translational regulation of RKIP in melanoma involves mir-21, one of the targets of NANOG and EMT. Some evidence suggests a direct correlation between mir-21 and RKIP levels as overexpression of RKIP in primary melanoma cultures is associated with a significant reduction of mir-21 and at the same time with lower cell mobility [68].

### 3.6. Clear Cells Renal Cell Carcinoma 

RKIP is primarily expressed in the proximal tubules of the kidney [73]. Hill B. et al. compared adjacent non-tumour kidney (ANK) tissue from 90 patients affected by renal cancer with 50 tissues from clear cell Renal Cell Carcinoma (ccRCC) and found a significant reduction of RKIP in the ccRCC cohort. The results obtained in the preliminary analysis were confirmed by TMA on approximately 600 renal tissue samples (45 ANK) versus 571 ccRCC, including 556 organ-confined tumours without metastasis and 15 carcinomas with metastasis.

The comparison between the expression levels of RKIP and the grade and stage of the tumours confirmed the direct correlation between the reduction of RKIP levels and tumour progression.

Of note, since ccRCC originates from epithelial cells of the proximal tubule [74,75], the progressive expansion of the tumour clone which replaces the normal architecture of the renal tubule leads to a progressive downregulation of RKIP as a consequence of loss of epithelial cells expressing RKIP. 

However, in vitro studies on ccRCC cell lines A498 and 786-0 suggested that the reduction of RKIP in tumour cells is at least in part mediated by the methylation of the RKIP promoter which represses its expression; On the contrary, the demethylation of the RKIP promoter increases its levels both as mRNA and protein. On the whole, the loss of RKIP has a functional impact on tumour cell proliferation and is involved in tumour progression and metastasisation; in fact, the overexpression of RKIP in ccRCC cellular models inhibits EMT and reduces invasion ability [73]. The role of pRKIP in renal cancer also needs to be clarified. Our group evaluated the expression levels of RKIP and pRKIP in both the tissue and urine of patients with ccRCC [32]. We carried out immunohistochemical staining of RKIP and p-RKIP proteins in tissue microarrays on specimens from normal human kidneys and patients with chronic kidney diseases and ccRCC. RKIP was markedly reduced in ccRCC compared to normal tissue and chronic kidney diseases thus emphasizing the concept that impairment of RKIP expression is directly linked to neoplastic transformation. Furthermore, we reported for the first time a total lack of pRKIP in ccRCC tissues and hypothesized that it may depend on the downregulation of RKIP which, in turn, leads to a drastic reduction of both the total protein and its phosphorylated form up to making it no more detectable in kidney tissues. In this context, pRKIP could constitute an early surrogate marker potentially useful for the early diagnosis of ccRCC as it allows for identification of the early stages of neoplastic transformation that correlate with the initial downregulation of RKIP.

**Table 1 cancers-14-05070-t001:** Main contribution of epigenetic, post-transcriptional and post-translational regulation of RKIP in the onset and progression of human cancers.

Human Cancers	RKIP Dysregulation	Ref.
** *Lung Cancer* **	RKIP mRNA downregulation correlates with poorer differentiation	[42]
	Lower RKIP protein level identify late stage NSCLC	[43]
	Increased miR-362 and miR 543 promotes metastases	[45]
	Increased miR-543 expression and RKIP downregulation involve impaired expression of lncRNAS like GATA6 RNA1	[48]
	pRKIP expression may represent an early biomarker of inflammation and RKIP downregulation	[52,53]
** *Colon Cancer* **	Reduced RKIP expression correlates with lower differentiation and predicts the development of metastases;	[56]
	The distribution of pRKIP between the nucleus and cytoplasm of cancer cells may constitute an early event in the process of colorectal cancer progression;	[31]
	Increased miR224 downregulates RKIP and activates RAS/MAPK signalling while reduced miR330 sustains EMT;	[57,59]
** *Breast Cancer* **	Lower levels of RKIP correlates with tumor size and grade	[62]
	miR-224 represses RKIP expression that, in turn, may contribute to tumorigenesis and metastatization by blocking the expression of the inhibitory miR-185	[30,61]
	Increase levels of pRKIP seems to exert a protective role against tumor progression and metastatization	[62]
** *Myeloid neoplasms (MNs)* ** ** *and Multiple Myeloma* **	miR-23a dependent downregulation of RKIP promotes metastases in AML	[65]
	MM tissue shows higher expression of RKIP and YY1 compared to normal bone marrow cells that, in turn, favour chemo/immuno-resistance;	[67]
	Increased pRKIP is the main mechanism of RKIP inactivation and tumor progression in MM	[66]
** *Melanoma* **	Early stage malignant melanoma shows reduced RKIP levels than benign lesions	[68]
	pRKIP is increased in non-metastatic melanoma compared to normal tissue and metastatic melanoma	[69]
	RKIP silencing correlates with higher expression of many oncogenes in melanoma cells	[70]
	Post-transcriptional regulation of RKIP in melanoma involves increased levels of mir-21 and NANOG	[72]
** *clear cells Renal Cell Carcinoma* **	RKIP is primarily expressed in proximal tubular cells of the kidney	[73]
	RKIP downregulation in ccRCC may depends in part by promoter methylation and in part by the destruction of tubular cells during tumor progression	[73,75]
	Reduction of RKIP expression correlates with the grade and stage in ccRCC	[73]
	pRKIP appears completely undetectable in both tissue and urine of ccRCC patients and it may represent a good candidate biomarker for early diagnosis	[32]

## 4. RKIP in Biofluids

A growing body of evidence clearly points out the central role of RKIP in the development and progression of many cancers. This has been possible by analysing various tumour tissues, however, these samples are less suitable for screening purposes since they can be obtained only by invasive procedures that cannot be justified in asymptomatic subjects. The introduction of RKIP into the group of new potential tumour biomarkers requires its validation in biological fluids commonly used in clinical practice. The feasibility of this application is still to be demonstrated; in fact, to date, only a few studies have evaluated this biomarker in biological fluids. Our group has been the first one to demonstrate, in ccRCC, that the expression levels of RKIP and pRKIP in urine reflected those of tumour tissues, thus laying the groundwork for the potential use of RKIP and pRKIP for diagnostic and prognostic screening of subjects at risk for this pathology [32,76]. 

We demonstrated that urinary RKIP was significantly less excreted in ccRCC patients than in healthy subjects and that a cut-off value of 10 ng/mg/g Pr/uCR at diagnosis was able to predict disease progression and death over time. Furthermore, we demonstrated that urinary RKIP is reduced also in chronic kidney disease but this non-cancer condition can be distinguished by evaluating urinary pRKIP. Thus, parallel analysis of urinary RKIP and pRKIP may provide a highly sensitive approach for early diagnosis and prognosis of ccRCC. 

Beyond this application, to date, there are no other works that have measured RKIP or its modified forms in the biological fluids of patients affected by other cancers. However, RKIP has been described in some biological fluids such as serum. Recent work by Bedri SK et al. [76] reported the usefulness of serum RKIP assessment for the monitoring of the treatment of multiple sclerosis. The analysis of 59 patients with multiple sclerosis, showed a significant correlation between the efficacy of treatment with natalizumab (humanized antibody targeting the α4β1 integrin) and the reduction of plasma RKIP levels. The authors suggest that RKIP may be a surrogate marker for inflammation as it is specifically reduced in subjects treated with a highly anti-inflammatory monoclonal antibody. Although this study is not correlated with any neoplasia, it nevertheless has the merit of having established for the first time the possibility of evaluating the assays of serum RKIP as a disease surrogate marker.

## 5. Conclusions and Perspectives

There is a growing body of literature that has investigated the role of RKIP in various neoplasms. Many of the regulatory mechanisms of this important player in the neoplastic process have been already clarified and revealed a predominant role of RKIP in the progression and metastasis of many cancers. At the same time, data on the role of post-translational changes of RKIP are still fragmentary but suggest that the evaluation of the phosphorylated form of RKIP may play an important role to clarify its function.

Data produced so far indicate that the role of RKIP phosphorylation as a positive or negative prognostic factor seems to be related to the type of neoplasm and probably to the microenvironment of each tumour type (Table 1). In particular, if the increase in the phosphorylated form of RKIP constitutes mainly an active mechanism of post-translational regulation, the reduction may depend not only on the action of specific phosphatases but also on the reduced expression of RKIP. It will therefore be necessary to carry out specific in vitro studies to clarify the concomitant role of epigenetic regulation and post-translational modifications on RKIP activation/inactivation in various tumour settings. 

At the same time, the usefulness of the assessment of RKIP in biofluids to allow its use as a surrogate biomarker is a potentially important but still less explored field. A potentially useful way to establish the usefulness of RKIP as a new tumour biomarker could be to evaluate its expression in various biological fluids in comparison with other biomarkers already used in clinical practice. This would allow us to evaluate the diagnostic sensitivity and specificity of RKIP and pRKIP with respect to other well-established tumour markers and to identify the most suitable biological fluid for each tumour. The greatest difficulty in achieving this goal in a short time undoubtedly lies in the capacity of cooperation between translational medicine scientists and clinicians. This is a significant challenge but certainly necessary to achieve this ambitious goal.

## Figures and Tables

**Figure 1 cancers-14-05070-f001:**
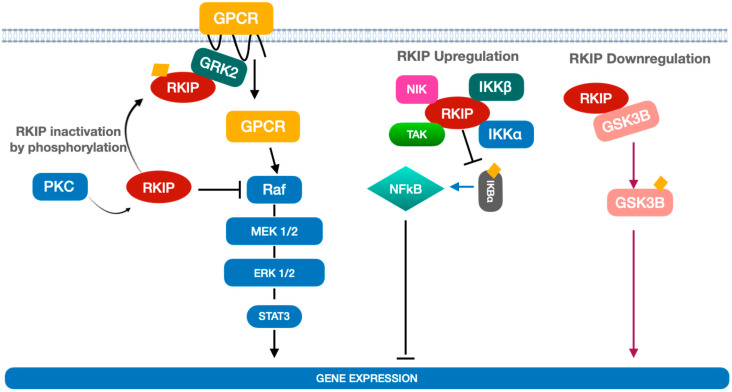
Mechanisms of RKIP regulation and their impact on biological processes.

**Figure 2 cancers-14-05070-f002:**
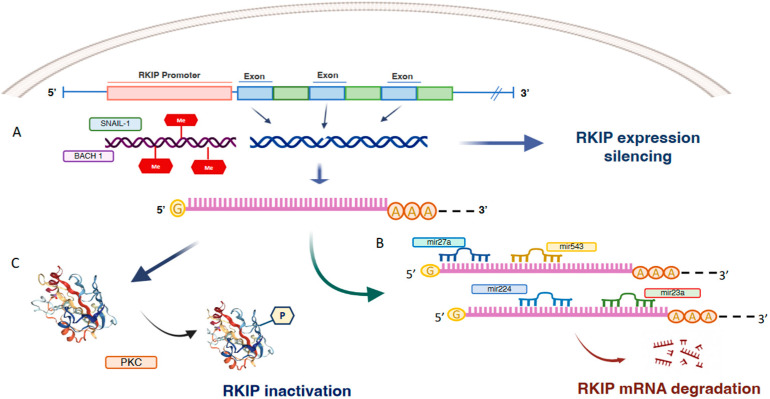
Transcriptional, post-transcriptional and post-translational mechanisms of RKIP inactivation. Schematic representation of the main regulatory mechanisms of RKIP. At transcriptional level, RKIP expression is inhibited by promoter hypermethylation and the binding of inhibitory transcription factors such as SNAIL-1 and BACH1 (**A**). Furthermore, in many cancer types RKIP mRNA is targeted by a number of micro RNAs that act as inhibitors of RKIP expression (**B**). Finally, post- translational regulation of RKIP implies RKIP phosphorylation by PKC that makes it inactive (**C**).

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
