# Peer review of "Understanding Mechanisms of RKIP Regulation to Improve the Development of New Diagnostic Tools"

_cancers, 2022, doi:10.3390/cancers14205070_

Round 1

Reviewer 1 Report

This review article summarizes the literature on the function of RKIP, its regulation, and its role in cancer.

I’d like to raise a few issues.

  • The abstract states that a special emphasis will be given to the post-translational modification of RKIP as opposed to other forms of regulation. The authors did not justify this choice nor showed that post-translational modification is more important than regulation at other levels. Furthermore, the majority of the regulatory links presented in the article are at the transcription or post-transcriptional level. Finally, the authors state regarding the post-translational modification that “its role is unexplored” (on page 6, last paragraph).
  • In their conclusions, the authors state that RKIP phosphorylation could serve as a positive or negative prognostic factor (page 9). The bulk of the text consists of studies that support the notion of RKIP as an antimetastatic whose expression is lost in cancer. Therefore, the statement is unsupported.
  • Despite attempting to cover the epigenetic regulation of RKIP, the authors do not reference or discuss a large portion of the literature that the role of transcription factors in controlling RKIP expression and their dysregulation in multiple types of cancer.
  • Many of the paragraphs are either too short or too long paragraphs. It makes comprehending and following the narrative very difficult. I suggest the authors restructure their content into balanced paragraphs and make use of subheadings when a section is too long.
  • The content in Table 1 would be better presented in a graphical format since it contains several signaling pathways.
  • The content in Figure 2 would be better presented as a table. A figure is probably not the most suitable way to present this amount of text.
  • The in-line referencing style (number, author, year) is confusing and inconsistent.

Author Response

We want to thank the reviewer for his careful review of the manuscript. Below is the point-by-point response to the reviewer's remarks

  • The abstract states that a special emphasis will be given to the post-translational modification of RKIP as opposed to other forms of regulation. The authors did not justify this choice nor showed that post-translational modification is more important than regulation at other levels. Furthermore, the majority of the regulatory links presented in the article are at the transcription or post-transcriptional level. Finally, the authors state regarding the post-translational modification that “its role is unexplored” (on page 6, last paragraph).

Based on his observations, which we share, we decided to rewrite the abstract by reporting in a more neutral way the impact of each regulatory mechanism (transcriptional, post-transcriptional, post-translational) on the expression of RKIP. Furthermore, in the abstract we highlighted that the role of RKIP has been assessed in detail only for tumors for which experimental data are available on each mechanism of regulation of expression and / or activity.

  • In their conclusions, the authors state that RKIP phosphorylation could serve as a positive or negative prognostic factor (page 9). The bulk of the text consists of studies that support the notion of RKIP as an antimetastatic whose expression is lost in cancer. Therefore, the statement is unsupported.

We thank the reviewer for this comment. However, we want to highlight that, in our opinion, the element of novelty of this work compared to the current literature lies precisely in the identification of the scientific works that have allowed to highlight the mechanisms of regulation of RKIP at the transcriptional, post-transcriptional and post-translational level. In this context, post-translational regulation also seems to have a weight, directly or indirectly, on the functionality of RKIP. A number of studies suggest prognostic role of pRKIP in several cancers. For example, RKIP phosphorylation alters (inactivating) the function of RKIP in multiple myeloma contributing to disease progression (Huerta-Yepez, S., 2014); The increased nuclear pRKIP may constitute an early event in the process of colorectal cancer progression (Cross-Knorr, S. et al., 2013;); Furthermore, increase levels of pRKIP seems to exert a protective role against tumor progression and metastatization in breast cancer (Al-Mulla, F. et al., 2013); Again, pRKIP is significantly increased in non-metastatic melanoma compared to normal tissue and metastatic melanoma (Cardile et al., 2013;). Finally, the phosphorylated form of RKIP is not detectable in the tissues and urine of patients with renal cancer already in the early stages therefore this marker could play a role in the early diagnosis of the disease (Papale, M. et al., 2017;). For the reasons set out and the scientific evidence reported, we believe that the sentence highlighted by the reviewer should not be removed from the text.

  • Despite attempting to cover the epigenetic regulation of RKIP, the authors do not reference or discuss a large portion of the literature that the role of transcription factors in controlling RKIP expression and their dysregulation in multiple types of cancer.

We thank the reviewer for this observation. In the revised version of the manuscript we have included an introductory paragraph on the role of transcription factors in the positive and negative regulation of RKIP expression. Furthermore, we have also introduced this concept in the subsections of the role of RKIP in each described tumor.

  • Many of the paragraphs are either too short or too long paragraphs. It makes comprehending and following the narrative very difficult. I suggest the authors restructure their content into balanced paragraphs and make use of subheadings when a section is too long.

We thank the reviewer for this suggestion. We have carefully revised the manuscript trying to make the reading and comprehension of the text smoother.

  • The content in Table 1 would be better presented in a graphical format since it contains several signaling pathways.

The contents of Table 1 have been reported in graphical form (new figure 1) as suggested by the reviewer.

  • The content in Figure 2 would be better presented as a table. A figure is probably not the most suitable way to present this amount of text.

The contents of Figure 2 have been presented in table (new table 1) as suggested by the reviewer.

  • The in-line referencing style (number, author, year) is confusing and inconsistent.

The references in the text have been changed according to the style of the magazine.

Reviewer 2 Report

Comments:

This review on RKIP provides a good descriptive base on the regulation mechanisms of this protein as well as its functions.

However, regarding the section on its involvement in cancer, only 4 types are mentioned (colon cancer, breast cancer, myeloma, and kidney cancer). The expression of this protein is high in tissues such as the skin or ovarian. And its relationship with malignant progression has been extensively studied in both kinds of cancers. However, these tumor types are not mentioned in this review.

Despite there are recent reviews on RKIP that provide information similar to that collected in this review, this one provides a novel section about the expression of RKIP in biofluids. However, as the authors themselves specify, there are hardly two works, one carried out by the authors. There may be a conflict of interest.

Suggestions:

1. Would it be possible to implement the information related to other types of tumors, such as melanoma, for example, in which the expression of RKIP has been studied extensively and whose management of patients needs improvement?

2. Rethink the title to better fit the content of the review, since although the regulatory mechanisms are well described, it is not well discussed in the results section how the development of new treatment tools could be improved. What would the targets be? And, regarding diagnostic tools, the use of a blood test, for example, to determine serum RKIP levels could be a good strategy to explore, although the studies that support this hypothesis are very limited.

Author Response

We want to thank the reviewer for his careful review of the manuscript. Below is the point-by-point response to the reviewer's remarks

  1. Would it be possible to implement the information related to other types of tumors, such as melanoma, for example, in which the expression of RKIP has been studied extensively and whose management of patients needs improvement?

We thank the reviewer for this suggestion. In the revised version of the MS, We have included also melanoma among the described tumors.

  1. Rethink the title to better fit the content of the review, since although the regulatory mechanisms are well described, it is not well discussed in the results section how the development of new treatment tools could be improved. What would the targets be? And, regarding diagnostic tools, the use of a blood test, for example, to determine serum RKIP levels could be a good strategy to explore, although the studies that support this hypothesis are very limited.

 We deleted the word "therapeutic" because, as correctly highlighted by the reviewer, this work describes the mechanisms of regulation of RKIP expression and function and their impact on the development and progression of various tumors. We left the word "diagnostic" in the title because, starting from the description of the various mechanisms of transcriptional, post-transcriptional and post-translational regulation, we tried to identify and discuss, based on scientific evidence, the usefulness of RKIP and pRKIP for the diagnosis and prognosis of various tumors. 

Round 2

Reviewer 1 Report

The authors have addressed most points raised in the initial review. In particular

  • The abstract was revised to reflect the contents of the review.
  • New contents were added to cover the role of transcription factors in controlling RKIP expression (see below).
  • Contents were appropriately presented as tables and figures.
  • The main text was revised for structure and referencing style

Regarding the role of transcription factors in controlling RKIP. I would suggest the authors also discuss the following studies (Disclaimer, I am a co-author on these studies)

  1. https://pubmed.ncbi.nlm.nih.gov/30115852/
  2. https://www.ncbi.nlm.nih.gov/pmc/articles/PMC8002422/
  3. https://pubmed.ncbi.nlm.nih.gov/34885208/

Reviewer 2 Report

The modifications made have greatly improved the original manuscript.  The drawings of the figures help to understand the text and more bibliography related to different cancers has been incorporated.  congratulations!